# Assessment of Knowledge, Practice and Guidelines towards the Novel COVID-19 among Eye Care Practitioners in Nigeria–A Survey-Based Study

**DOI:** 10.3390/ijerph17145141

**Published:** 2020-07-16

**Authors:** Bernadine Ekpenyong, Chukwuemeka J. Obinwanne, Godwin Ovenseri-Ogbomo, Kelechukwu Ahaiwe, Okonokhua O. Lewis, Damian C. Echendu, Uchechukwu L. Osuagwu

**Affiliations:** 1Department of Public Health, Faculty of Allied Medical Sciences, College of Medical Sciences, University of Calabar, Calabar 540271, Nigeria; bekpenyong@unical.edu.ng; 2African Eye and Public Health Research Initiative, African Vision Research Institute, Discipline of Optometry, University of KwaZulu-Natal, Durban 3629, South Africa; godwin.ovenseri-ogbomo@uniben.edu; 3Cornea and Contact Lens Department, De-Lens Ophthalmics Family Eye and Vision Care Center, Abuja 900281, Nigeria; obinwannejr@gmail.com; 4Department of Optometry, College of Applied Medical Sciences, Qassim University, Buraidah 51452, Saudi Arabia; 5Department of Optometry, Faculty of Life Sciences, University of Benin, Benin City 300283, Nigeria; 6Optometry Unit, Department of Ophthalmology, University of Calabar Teaching Hospital, Calabar 540242, Nigeria; kcahaiwe@yahoo.com; 7Calabar Diabetic Retinopathy Screening Service, University of Calabar Teaching Hospital, Calabar 540242, Nigeria; 8Department of Medical and Diagnostics, Health and Human Services Secretariat, Federal Capital Territory Administration, Abuja 900247, Nigeria; imuse288@gmail.com; 9Department of Ophthalmology, State House Medical Centre, Abuja 900001, Nigeria; damian@delensng.com; 10Ocular Health and Behavioral Optometry Department, De-Lens Ophthalmics, Abuja 900004, Nigeria; 11Diabetes, Obesity and Metabolism Translational Research Unit, Western Sydney University, Campbelltown 2560, New South Wales, Australia

**Keywords:** coronavirus, personal protective equipment, essential service, pandemic, eye care practitioners

## Abstract

The aim of this study was to explore knowledge, practice of risk and guidelines of the novel corona virus disease (COVID-19) infection among the eye care practitioners and the potential associated factors. A cross-sectional self-administered online survey was distributed via emails and social media networks between 2nd and 18th May 2020 corresponding to the week of the lockdown in Nigeria to eye care practitioners (ECPs). Data for 823 respondents were analyzed. Knowledge and risk practice were categorized as binary outcome and univariate and multivariate linear regression were used to examine the associated factors. The mean score for COVID-19-related knowledge of public health guidelines was high and varied across the ECPs. Ophthalmic Nurses, Ophthalmologists and Optometrists showed higher COVID-19-related knowledge than other ECPs (*p* < 0.001), particularly those working in the private sector. More than 50% of ECPs stated they provided essential services during the COVID-19 lockdown via physical consultation, particularly the Ophthalmologists. Most respondents reported that the guidelines provided by their Association were useful but expressed their lack of confidence in attending to patients during and after the COVID-19 lockdown. Compared to other ECPs in Nigeria, more Ophthalmic Nurses received training in the use of Personal Protective Equipment (PPE). This survey is the first to assess knowledge, attitudes and practice in response to the COVID-19 pandemic in Nigeria. ECPs in Nigeria displayed good knowledge about COVID-19 and provided eye care services during the COVID-19 lockdown in Nigeria, despite the majority not receiving any training on the use of PPEs with concerns over attending to patients. There is need for the government to strengthen health systems by improving and extending training on standard infection prevention and control measures to ECPs for effective control of the pandemic and in the future as essential health workers.

## 1. Introduction

The emergence of the novel coronavirus disease in 2019 (COVID-19) in December 2019 in the city of Wuhan, the Chinese province of Hubei city, halted the ever-busy human society and threatened every nation [1]. A completely different type of acute pneumonia [2] which had close resemblance to the previous Middle East respiratory syndrome (MERS) and Severe acute respiratory syndrome (SARS) viruses but appeared to be much more lethal than the two was reported [3]. The infection soon became a cause of concern with the World Health Organization, declaring the rapid spread of cases of COVID-19 a pandemic on 11th March, 2020 and recommended that a globally coordinated effort was needed to fight the pandemic [4]. While there is currently no vaccine for COVID-19 [5], the symptoms can include fever, flu-like symptoms such as a cough, sore throat and fatigue and/or shortness of breath, diarrhea, nausea and vomiting [6]. The risk of death in COVID-19-infected individuals increases with older age, presence of hypertension, diabetes and coronary heart diseases [7]. There are also reports of conjunctivitis and transmission of the virus by aerosol contact with conjunctiva [8] with some uncertainty as to whether the virus is evident in human tears [1].

On the 28th of January 2020, sub-Saharan Africa’s first confirmed case of COVID-19 was announced In Nigeria. This led to the activation of the country’s National Coronavirus Emergency Operation Centre by the government. During to the Ebola outbreak of 2014, of the 15,000 confirmed cases, there were over 9000 suspected cases in West Africa, but this was controlled in just 92 days [9]. Currently, the control of COVID-19 is becoming challenging for the Nigerian government despite the mobilization of resources and manpower by the Nigeria Centre for Disease Control NCDC [9,10]. There are about 16,658 confirmed cases of COVID-19 and 424 lost lives of humans from the infection (16 June 2020). The majority of the cases are in the former capital city of Lagos (7319 cases, 82 deaths), Federal Capital city of Abuja (1264 cases, 26 deaths) and Kano (1158 cases, 50 deaths) [10].

As the country continues to experience steady increase in the number of confirmed cases [10], the different levels of government have taken proactive steps to curtail the spread of coronavirus throughout the country. Movements were restricted within and between states, and the society observed a partial lockdown in response to the pandemic. Current evidence suggests that the implementation of outbreak response strategies for COVID-19 can limit the disease. However, these situational responses affect businesses including their interactions with relevant regulators/professional bodies causing the Government to respond through the Nigerian National Assembly’s Emergency Stimulus Bill, the Central Bank of Nigeria’s policy measure which dedicated its credit facility to develop the healthcare sector [11].

Unlike some businesses and occupations considered as essential services, eye care professions (ECP) discontinued operations during the lockdown denying many patients—particularly those in need of emergency care or receiving routine injections for management of blinding eye diseases such as diabetes macular edema—access to eye care. ECPs may be susceptible to infection due to close patient proximity during examination such as slit lamp examination, applanation tonometry and the potential contamination of instruments [12]; however, medical visits related to systemic and ocular disease or injury where there is significant risk of permanent vision loss because of any postponement of care, as determined by the treating ECP, are considered essential visits [13]. Other conditions considered by ECPs as essential services have been summarized in Table 1. Additionally, the same groups burdened by COVID-19 complications could also suffer more vision problems including individuals with hypertension, respiratory conditions, and heart disease and the elderly [14]. Patients who have lost or broken their glasses or contact lenses with consideration given to prescription needs and level of disability without correction are considered as essential services [13]. There are also concerns existing around the pandemic with various reports from news outlets and social media reporting how best to limit the chance of infection, with significant amounts of misinformation and speculation [5] which many patients may request clarification from their ECPs to keep them safe through this period.

The aim of this study was to assess knowledge and practice of COVID-19 exposure risk among ECPs as well as understand their confidence in current Federal Ministry of Health (FMoH) guidelines for identifying possible COVID-19 cases, knowledge of Personal Protective Equipment (PPE) recommendations and training in its usage when managing such cases. The impact of COVID-19 lockdown among practitioners was also assessed. This survey is among the first to assess knowledge level, practice of risk and awareness of the guidelines for consulting patients at risk or confirmed cases of COVID-19 in Nigeria incorporating responses from all tiers of ECPs in Nigeria. The findings will also provide first evidence on ECPs’ knowledge of COVID-19 in Nigeria. This will help to reduce their risk, and that of their family, of contracting the virus, reduce morbidity and mortality associated with being infected. Evidence from the study can also be used to implement emergency policies to counter the spread and impact of a similar outbreak in future. The study will provide clarity on the essential nature of ECPs services to help policy making in future outbreaks.

## 2. Materials and Methods

### 2.1. Study Population

This study on the knowledge, practice, impact and guideline on COVID-19 was conducted among eye care practitioners in Nigeria. According to The World Bank Group (2019), Nigeria has an estimated population of 195,874,740 people. Majority of eye care service practitioners are located in the cities [15]. Nigeria is home to 7000 registered optometrists [16], about 300 ophthalmologists [17], 2000 ophthalmic nurses [18] and 941 dispensing opticians [16].

All eye care practitioners practicing in Nigeria have overlapping roles without distinct borders. Ophthalmologists undergo a minimum of four (4) years postgraduate training after a medical degree and provide surgical as well as medical eye care [19]. Optometry is a licensed professional program completed in a minimum of six (6) years leading to the award of Doctor in Optometry (OD) which empowers Optometrists to provide general eye care including treating eye diseases, refractive errors, low vision and contact lenses [16]. An Ophthalmic nurse has a one-year post-basic nursing training in eye care and work with other ECPs to engage in blindness prevention activities and care for patients for ocular surgeries. Dispensing opticians obtain a three-year National Diploma and work in optical laboratories to interpret and dispense optical prescriptions [20].

A self-administered questionnaire developed and used previously for ECPs [21] was modified and pre-tested to ensure that it was suitable for use in Nigeria. The initial survey was piloted among 10 Optometrists who were not part of the study team and did not participate in the final survey to ensure clarity and understanding as well as to determine the duration for completing the questionnaire prior to disseminating them.

### 2.2. Ethics

The study adhered to the principles of the 1967 Helsinki declaration (WMA, 2013) and the protocol was approved by the Human Research Ethics Committee of the Cross River State Ministry of Health, Nigeria (Ref #: CRSMOH/RP/REC/2020/116). Participation was anonymous and voluntary. Informed consent was obtained from all participants prior to commencement of the study and after the study protocol has been explained. Participants consented to voluntarily participate in this study by answering either a ‘yes’ or ‘no’ to the question inquiring whether they voluntarily agree to participate in the survey. A ‘no’ response meant that the participants could not progress to answering the survey questions and were excluded from the study.

### 2.3. Sample Size Determination and Sampling Procedure

The required sample size for this study was determined using a single population proportion formula given as:(1)$$n=\frac{Z^{2}pq}{d^{2}}=\frac{1.96^{2}x0.50\;x0.50}{0.04^{2}}=600$$

In the absence of similar studies in Nigeria, the study assumed a proportion of 50% of the population and used a desired precision of 4% and 95% confidence level for a two-sided test. To make up for non-response rate of 25%, the sample size was determined to be 800 persons, which was adequate to detect statistical differences in the analysis of online cross-sectional study on COVID-19 among ECPs in Nigeria. Respondents were proportionately determined across the 4 categories of ECPs. A self-administered anonymous online survey was administered using convenience sampling technique, on a first-come bases until the required number was obtained within the one-month duration of the survey. A total of 823 questionnaires were fully completed and retrieved in the estimated proportions for the different categories of ECPs except for Ophthalmic Nurses where we got less than the required sample (Ophthalmologists [*n =* 66], Optometrists [*n =* 598], Ophthalmic nurses [*n =* 48] and Dispensing Opticians [*n =* 111] ).

### 2.4. Procedure

The survey was created in survey monkey and disseminated to registered ECPs in Nigeria including Optometrists, Ophthalmologists, Opticians, Ophthalmic nurses, and Ophthalmic technicians between 2nd and 18th May 2020. Distribution was through the administrative heads of the various professional bodies including the Ophthalmological Society of Nigeria (OSN), Nigerian Optometric Association (NOA), Nigeria Ophthalmic Nurses Association (NONA) and Association of Nigerian Dispensing Opticians (ANDO) and individually. A link to the online survey was disseminated via the emails and social media platforms (Facebook and WhatsApp) of the different professional organizations. Survey link remained active from 2 May to 18 May 2020, within which time participants completed the survey. The practitioners did not receive incentives for participating in the study and were not under any obligation to complete the survey.

Participants included ECPs who were currently registered to provide clinical services at different levels of eye care within Nigeria at the time of the study. Responses from non-ECPs, non- Nigerians, ECPs practicing outside Nigeria, and non-practicing practitioners were excluded from the analysis.

### 2.5. Instrument for Data Collection

The survey tool was shown in Appendix A and consisted of 36 items divided into five sections (demographic characteristics, knowledge, practice of risk of contracting the infection, impact and guidance) utilizing closed-ended questions and a four point ‘Likert-type scale’ to score participants’ responses. The responses ranged from ‘yes’ (score ‘1′) to ‘no’ (score ‘-1′). A ‘not sure’ response was scored as ‘zero’. For responses utilizing Likert scale, the scores ranged from ‘3′ for ‘extremely confident’ to ‘1′ for confident and ‘-1′ was scored for ‘not-confident’

The impact of COVID-19 pandemic on practitioners, their family members and practices, including questions on their confidence in the current FMoH guidelines for identifying possible COVID-19 cases, their knowledge of Personal Protective Equipment (PPE) recommendations, and training in its usage during consultation were assessed.

### 2.6. Independent and Dependent Variables

The explanatory (independent) variable included basic characteristics and explanatory factors including gender, age in categories, region of practice, level of education, marital, employment and religion status, type of ECP, practice setting and practice years.

The dependent variables in the regression analysis was knowledge relating to COVID-19. The total score ranged from 1 to 9. The scores were derived from questions inquiring on ‘whether the participants knew the occupation classified as ‘Essential work’ by the Ministry of Health during the COVID-19 lockdown’, if ECPs could correctly identify from a list of nine items, the recommended PPEs by the NCDC in preventing COVID-19 transmission, during consultation of confirmed/suspected cases for health care workers?

### 2.7. Statistical Analysis

Descriptive statistics and Multivariable analysis were performed to demonstrate the outline of the findings of this study and sample characteristics. The responses were presented descriptively in tables. First, the entire cohort—men and women— was analyzed —to determine the knowledge towards COVID-19. Then, chi-square tests were used to examine the variability in responses by gender, for the different ECPs, concerning the knowledge, practice and understanding of the guidelines of the FMoH. The variability in responses between ECPs from the different specialties concerning their understanding of guidelines was also assessed. Univariate linear regression analysis was calculated in order to assess the unadjusted coefficient. All confounding variables with a *p* value < 0.20 were retained and used to build a multivariable linear regression model. A manual stepwise backwards model was used to estimate the adjusted estimate for independent variables and to determine factors associated with KAP scores towards COVID-19. A *p*-value ≤ 0.05 was considered statistically significant and we checked homogeneity of variance and multicollinearity using Variance Inflation Factors (VIF). All statistical analyses were carried out using the Statistical Program for Social Sciences, version 25.0 (SPSS Inc, Chicago, Illinois, USA).

## 3. Results

### 3.1. Demographic Profile of the Respondents

A total of 823 respondents (males, *n* = 374, 45.4%, females *n =* 449, 54.6%) aged 21–72 years (mean age ± SD, 38 ± 10 years) completed the online questionnaire. About 84.3% were aged less than 50 years and male respondents were significantly older than the females (39 ± 10 years, 95% CI 38–39.7 versus 37 ± 10 years, 95% CI 36.3–38.2; *p* = 0.033). Table 2 presents the demographic characteristics of the respondents including their employment status and years of practice.

### 3.2. Knowledge Relating to COVID-19

The total knowledge score relating to COVID-19 ranged from 1 to 9 with a mean score of 6.98 ± 2.00. Figure 1 shows the mean knowledge score for each eye care profession in the survey. There was a significant difference in the mean knowledge score between the professions (one way analysis of variance, *p* < 0.0001) with post hoc analysis revealing that the differences was only when Ophthalmic nurses (7.71 ± 1.81), Optometrists, Ophthalmologists (7.10 ± 1.85 and 7.39 ± 2.08, respectively) were compared with the Opticians (5.77 ± 2.34, *p* < 0.0001) who had the least knowledge of COVID-19 transmission. No other multiple comparison showed significant difference.

In the multivariable analysis, we found that, after adjusting for all cofounders in the final model, eye care profession (job title) was the only factor associated with knowledge of risk towards COVID-19 (adjusted coefficient, −0.182, 95% Confidence Interval −0.601, −0.22; *p* < 0.0001) (Table 3).

### 3.3. Perception of Risk of Contracting COVID-19 During the Lockdown Period

Table 4 shows the opinion of ECPs with respect to COVID-19 during the lockdown. Over 70% of the subjects reported lack of confidence in the guideline of the Federal Ministry of Health did not consider eye care workers as “Essential workers” during the lockdown. Notwithstanding, 43.2% were either not so confident or not at all confident attending to any patient during the lockdown while 54.6% also reported they were not so confident or not all confident attending to COVID-19 patient or those at risk of COVID-19. When questioned about their level of confident attending to patients after the lockdown, 26.3% of eye care professionals reported lack of confident attending to patients even after the lockdown is over and for majority of the practitioners (90%), COVID-19 will change the way the deliver eye care service in their practice.

The results also revealed that a high proportion of eye care professionals provided eye care services to patients during the lockdown (Figure 2) with more Ophthalmologists and an equal proportion of Optometrists and Ophthalmic Nurses providing services. Of the various means of consultation during the lockdown (Figure 2), it can be seen that many Ophthalmologists (73%), Optometrist and Ophthalmic nurses (65% and 62%, respectively) did so via physical consultations in the clinic. More Optometrist than Ophthalmologist (10.4% vs. 6.1%) utilized videoconferencing to provide this much-needed service during the lockdown while consultation over the phone, social media were also utilized by ECPs during the lockdown (Figure 2).

### 3.4. Practice of Professional Guidelines During COVID-19

Compared to other practitioners, a significant higher percentage of optometrists reported that their professional association provided information on guidelines during COVID-19 (Figure 3). For over 80% of the respondents from each eye care profession, the guidelines were useful and regarding the use of personal protective equipment (PPE), less than 40% of each eye care professionals received training on the use of PPE in the control of COVID-19. Slightly more ophthalmic nurses (28.9%) received training on PPE compared to the ophthalmologists (14.0%) but this was at borderline significance (*p* = 0.056) (Figure 3).

## 4. Discussion

This is the first study to assess the knowledge, attitude and guidelines of all tiers of ECPs regarding the Public Health initiatives for the novel coronavirus (COVID-19) in Nigeria. The study found that knowledge about COVID-19 preventive guidelines was high among ECPs and Ophthalmic nurses, Ophthalmologists and Optometrists were significantly more knowledgeable compared to Opticians. The majority of the ECPs did not receive training on the proper use of PPEs despite a significant proportion stating that they attended to patients during the lockdown period. Although the majority of the ECPs felt that their professional Association provided some useful information on guidelines during the pandemic, this was considered grossly inadequate for many of the Ophthalmologists and Ophthalmic nurses. More than half of the ECPs expressed lack of confidence in caring for patients at risk of COVID-19 and, for more than a quarter of them, this will continue even after the lockdown is over.

Similarly high COVID-19-related knowledge was reported in the general Nigerian population [22], and that of the Chinese population [23] as well as those of the health care practitioners [14] but an earlier survey found a lack of understanding of the Public Health guidelines related to COVID-19 among ECPs in the UK. The study included 100 ECPs (ophthalmologists, optometrists, ophthalmic nurses and healthcare assistants) [21]. Compared to the UK study, the present study found high knowledge scores among respondents and this difference may be related to timing of both studies as the time lag may have allowed for the respondents in the present study to learn more about COVID-19 and, as such, demonstrated higher knowledge scores. At the time of the UK study, the coronavirus outbreak had just been designated a pandemic by the WHO [4], although the first confirmed case was reported in the UK on 29 January 2020.

The significant association found between COVID-19-related knowledge and the category of ECP may be attributed to the Ophthalmic Nurses having more training on PPEs than other ECPs, which may have translated to the higher knowledge scores. Although the Nigerian Federal Ministry of Health do not consider ECPs as essential workers, a large proportion of the respondents disagreed with this and more than half confirmed that they provided emergency eye care services via physical examination of patients during the lockdown. This finding suggests the need to consider the inclusion of ECPs as part of the essential healthcare team since ocular emergencies can occur at any time and viral conjunctivitis may be a symptom of COVID-19 [16,24].

Several guidelines to limit the risk of infection and help ECPs safely provide eye care services have been published by the Ophthalmic Associations, Societies and Researchers during the pandemic [10,12,16,25,26,27,28,29,30]. This is vital as several procedures involve the practitioner to be in close proximity to patients and as such proper use of PPE is essential. A survey of Optometrists and Opticians conducted in Austria, Germany and Switzerland reported that over 50% of the ECPs planned to wear masks during refraction, contact lens fitting and practiced hand washing and disinfection before performing procedures [31]. However, training in the use of PPE is important to avoid the ECP being infected. The finding that majority of ECPs did not receive any training on proper use of PPEs, was concerning and potentially dangerous, as it puts the practitioner at high risk of contracting COVID-19 [32,33].

An interesting finding of this study was the increased use of telemedicine for delivering eye care services during the COVID-19 pandemic, although only a few utilized this service. There is need for education on the methods of delivering this service and the associated benefits for ECPs in Nigeria. In addition, the fact that majority of the participants in this study were Optometrist may be a reflection of the higher number of registered Optometrists compared to Ophthalmologists and the fact that most of them are practicing in urban centers [34].

This study has some limitations. Firstly, the majority of the respondents were practicing in urban areas and their responses may not represent that of ECPs practicing in rural areas. Secondly, the low number of responses from ophthalmic nurses was lower than estimated from their registry, and this may affect the responses obtained from the group. Future studies should consider other ways of reaching this subgroup as their knowledge and practice as front-line workers is important. In addition, further studies are needed to investigate the knowledge and preparedness of ECPs in rural settings to provide service during the COVID-19 pandemic in Nigeria. Despite these limitations, this study is strengthened by the larger sample size compared to a previous study [21]. Another strength of this study was the representation of the opinions of all tiers of ECPs who are involved in the delivery of eye care services during the lockdown in Nigeria. In addition, the study was the first to provide evidence on knowledge, practice and guidelines of African ECPs during a pandemic. It identified major gaps in the ability of the ECPs to continue providing care during and after the pandemic which, if not addressed, might put the ECPs and their patients at risk of contracting the virus infection during consultation. Addressing these gaps is important to build confidence among ECPs and their patients during a pandemic and, more so, as most African countries prepare for a possible second wave of the virus.

## 5. Conclusions

This study demonstrated that ECPs in Nigeria were knowledgeable about COVID-19 and readily explored several avenues to serve the Nigerian population during the COVID-19 lockdown. However, the ECPs reported lack of confidence on the non-inclusion of eye care workers as essential in the government guidelines for the control of this pandemic, which places them at increased risk. Therefore, to ensure that ECPs continue to provide the needed services during the pandemic or similar events, there is need for training on the proper use of PPE and recognition as essential worker; this will, in turn, boost their confidence when attending to patients even after the lockdown. The Nigerian government need to strengthen health systems by improving and extending training on standard infection prevention and control measures for effective control of the pandemic.

## Figures and Tables

**Figure 1 ijerph-17-05141-f001:**
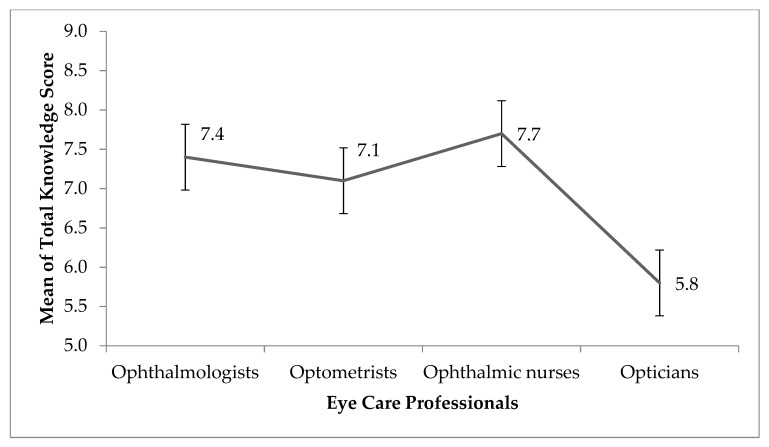
The mean knowledge score for each eye care profession in the survey. Error bars represent standard error of the mean.

**Figure 2 ijerph-17-05141-f002:**
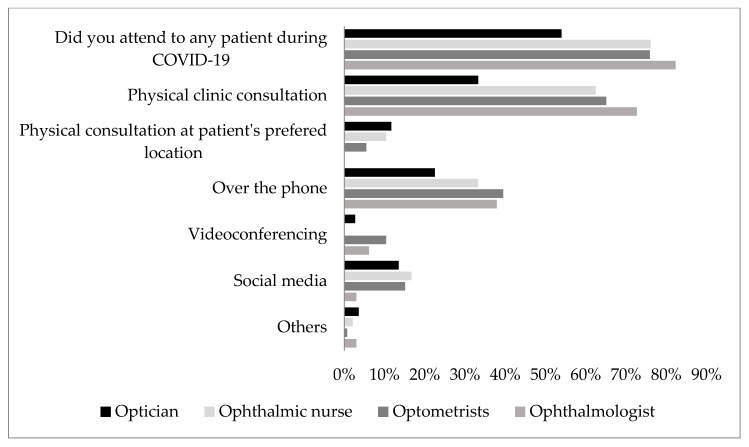
Provision of eye care services and the methods employed for the purpose by respondents during the novel coronavirus disease 2019 (COVID-19) lockdown.

**Figure 3 ijerph-17-05141-f003:**
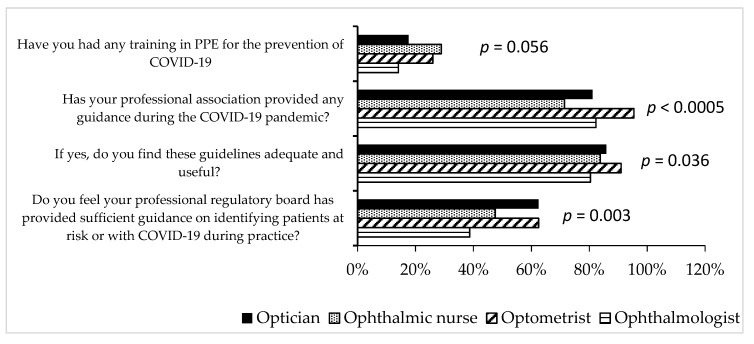
Practice of professional guidelines of respondents during the novel coronavirus disease 2019 (COVID-19) lockdown. PPE = Personal Protective Equipment.

**Table 1 ijerph-17-05141-t001:** Examples of essential care requiring emergency office visit.

Referral of patient from emergency department	House Price Index analysis of 2016 Healthcare Cost and Utilization Project data showed that 1% of all visits to the United States of America emergency department units were for eye-related encounters and that 98.9% of those eye-related encounters were treat and release that could be taken care of by doctors of optometry in their offices.
Trauma reported by patient	Blunt force, sharp object or foreign body or chemical to an eye; followed by pain, photophobia, sustained flashes of light, metamorphopsia or visual field loss.
Eye pain report by patient	Unexplained eye pain that cannot be resolved by virtual methods. This would include, but not limited to, acute angle closure glaucoma and corneal compromise (e.g., includes pain associated with contact lens wear and not resolvable after discontinuing contact lens wear).
Vision loss report by patient	Acute or gradual with or without pain, sudden onset blurred vision, color desaturation. Acute retinal arterial ischemia, including vascular transient monocular vision loss and branch retinal artery occlusion and central retinal arterial occlusions, are ocular and systemic emergencies requiring immediate diagnosis and treatment.
Double vision reported by patient	New onset.
Dropping of eyelid as reported by patient	Acute or sudden.
Flashes or floaters reported by patient with or without pain	New onset.

Source: American Optometry Association. Available at: https://www.aoa.org/coronavirus/health-policy-institute-covid-19/doctors-of-optometry-essential-care-guidelines-for-covid-19-pandemic.

**Table 2 ijerph-17-05141-t002:** Demographic profile of respondents.

Variables	Frequency (%)
*n* (%)	823 (100)
Age category (years)	
20–34	368 (44.7)
35–49	326 (39.6)
50+	129 (15.7)
Sex	823 (100.0)
Males	374 (45.4)
Females	449 (54.6)
Region of practice	820 (100.0)
Eastern Region	256 (31.2)
Western Region	246 (30.0)
Northern Region	211 (25.8)
Southern Region	107 (13.0)
Marital Status	823 (100.0)
Married	565 (68.7)
Not married	258 (31.3)
Highest level of education	823 (100.0)
Postgraduate Degree (Fellowship/Masters/PhD)	171 (20.5)
Bachelor’s degree	557 (67.7)
National Diploma	95 (11.5)
Eye care profession	823 (100.0)
Ophthalmologists	66 (8.0)
Optometrists	598 (72.7)
Ophthalmic nurses	48 (5.8)
Opticians	111 (13.5)
Religion	823 (100.0)
Christianity	764 (92.8)
Others	59 (7.2)
Practice setting	823 (100.0)
Public hospital/service	394 (47.9)
Private clinic/optical shop	429 (52.1)
Employment status	823 (100.0)
Self employed	178 (21.6)
Private employee	229 (27.8)
Government employee	382 (46.4)
Unemployed	34 (4.1)
Years of practice	822 (100.0)
1–12	560 (68.1)
13–24	156 (19.0)
25+	106 (12.9)

**Table 3 ijerph-17-05141-t003:** Multiple regression of factors associated with knowledge related to COVID-19 among eye care professionals in Nigeria during the lockdown.

Variable	Unadjusted Coefficient	Adjusted Coefficient	*p*-Value	95% CI of Adjusted Coefficient	
Age group ( years) (50+ = Reference)					
20–34	−0.016	0.984	0.975	0.357	2.710
35–49	−0.164	0.849	0.700	0.368	1.958
Marital status (Not married = Reference)					
Married	0.234	1.263	0.411	0.724	2.204
Religion (others = Reference)					
Christian	−0.628	0.534	0.166	0.219	1.297
Highest Educational Qualification (National Diploma = Reference)					
University degree (Bachelors/Doctor of Optometry/Professional degree)	0.716	2.046	0.238	0.623	6.721
Fellowship, Postgraduate degree and PhD	0.419	1.520	0.520	0.424	5.448
Job title (Optician = Reference)					
Ophthalmologist	−2.705*	0.067	0.001	0.014	0.323
Optometrist	−2.038*	0.130	0.004	0.032	0.527
Ophthalmic nurse	−2.623*	0.073	0.000	0.018	0.290
Place of work (Private hospital/clinic = Reference)					
Public hospital	−1.425 *	0.241	0.039	0.062	0.931
Employment status (Unemployed = Reference)					
Self employed	−0.556	0.574	0.488	0.119	2.758
Government employee	0.953	2.594	0.287	0.448	15.014
Private employee	−0.219	0.803	0.779	0.174	3.701
Years of practice (25+ = Reference)					
1–12	−0.134	0.875	0.787	0.331	2.308
13–24	−0.094	0.910	0.833	0.379	2.184

Dependent variable = Total knowledge score. * = statistical significance. CI = Confidence interval.

**Table 4 ijerph-17-05141-t004:** Practice of respondents during the lockdown.

Practice	Frequency (%)
How confident/informed do you feel in the Federal Ministry of Health guidelines that currently do not consider Eye care practitioners as ‘Essential workers’?	767 (100.0)
Extremely confident	43 (5.6)
Very Confident	79 (10.3)
Somewhat confident	105 (13.7)
Not so confident	227 (29.6)
Not at all confident	313 (40.8)
During the corona virus disease 2019 (COVID-19) lockdown, how confident do you feel attending to any patient?	769 (100.0)
Extremely confident	42 (5.5)
Very Confident	151 (19.6)
Somewhat confident	244 (31.7)
Not so confident	269 (35.0)
Not at all confident	(8.2)
How confident do you feel attending to a patient with or at risk of COVID-19?	768 (100.0)
Extremely confident	26 (3.4)
Very Confident	103 (13.4)
Somewhat confident	208 (27.1)
Not so confident	263 (34.2)
Not at all confident	168 (20.4)
After the lockdown, how confident would you feel attending to any patient?	770 (100.0)
Extremely confident	87 (11.3)
Very Confident	202 (26.2)
Somewhat confident	279 (36.2)
Not so confident	166 (21.6)
Not at all confident	36 (4.7)
How much would COVID -19 change the way you practice?	771 (100.0)
Very much	543 (70.4)
Moderately	179 (23.2)
Very little	35 (4.5)
Not at all	14 (1.8)

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
