# Peer review of "Assessment of Knowledge, Practice and Guidelines towards the Novel COVID-19 among Eye Care Practitioners in Nigeria–A Survey-Based Study"

_ijerph, 2020, doi:10.3390/ijerph17145141_

Round 1
Reviewer 1 Report
The paper is about a very important topic for the whole world these days. Control of this health emergency is of great importance due to the economic and health repercussions that COVID-19 brings to the population. The article should improve in the following aspects.
1. In the introduction the authors should improve the contribution of this article in the literature. Why is this paper important? Will it help to better control the emergency? Would the paper help to reduce deaths of people?
2. The authors speak in the abstract of a logistic regression. I don't see the model description. There is no explanation of the independent and dependent variables.
3. If there is already a survey, I suggest using the appropriate variables and the logistic model to achieve the objective of the paper.
4. In the model, only one significant coefficient is presented. It could be said that there is a bad specification of the model.
5. I suggest that the descriptive statistical information presented be used to estimate the model and interpret it in terms of probability.
6. Improve the conclusions.
Author Response
Response to Reviewers comments
We wish to thank the editor and the reviewers for their very useful comment which has helped to improve the quality of the manuscript. We have responded to every comment to the best of our ability and happy to provide further clarification if need be. In addition, the manuscript has been reviewed by a native English speaker and changes have been highlighted in red fonts across the manuscript.
Reviewer 1
The paper is about a very important topic for the whole world these days. Control of this health emergency is of great importance due to the economic and health repercussions that COVID-19 brings to the population. The article should improve in the following aspects.
1. In the introduction the authors should improve the contribution of this article in the literature. Why is this paper important? Will it help to better control the emergency? Would the paper help to reduce deaths of people?
Reply
The introduction has now been improved to include its contribution to the existing literature, importance and significance in reducing death.
The relevant section reads
This survey is among the first to assess knowledge level, practice of risk and awareness of the guidelines for consulting patients at risk or confirmed cases of COVID-19 in Nigeria incorporating responses from all tiers of ECPs in Nigeria. The findings will also provide first evidence on ECPs’. This will help to reduce their risk, and that of their family, of contracting the virus, reduce morbidity and mortality associated with being infected. Evidence from the study can also be used to implement emergency policies to counter the spread and impact of similar outbreak in future. The study will provide clarity on the essential nature of ECPs services to help policy making in future outbreaks.
The authors speak in the abstract of a logistic regression. I don't see the model description. There is no explanation of the independent and dependent variables.
Response:
We used linear regression and not logistic regression. The dependent and independent variables have now been described. See section 2.6
If there is already a survey, I suggest using the appropriate variables and the logistic model to achieve the objective of the paper.
Response:
Thanks for the comment. We have now presented the appropriate variables used in the linear regression analysis in Table 3
In the model, only one significant coefficient is presented. It could be said that there is a bad specification of the model.
Response:
We found two significant variables following adjustments for all potential cofounders. The abstract and result has also been revised to reflect this more clearly.
One of the sections in the abstract now reads:
The mean score for COVID-19 related knowledge of public health guideline was high and varied across the ECPs. Ophthalmic Nurses, Ophthalmologists and Optometrists showed higher COVID-19 related knowledge than other ECPs (p<0.001) particularly those working in private sector.
This method is similar to several other studies (Abir, et al 2020 doi: 10.20944/preprints202007.0054.v1) and provides a robust method of analysis. The sample size was also adequate indicating that our results are reliable and can be generalizable largely within the Nigerian community of ECPs.
I suggest that the descriptive statistical information presented be used to estimate the model and interpret it in terms of probability.
Response: This has been used to estimate the model used in the linear regression. The results are coefficients and appropriate interpretation has been followed.
Improve the conclusions.
Response: The conclusion has been improved. It now reads like so:
This study demonstrated that ECPs in Nigeria were knowledgeable about COVID-19 and readily explored several avenues to serve the Nigerian population during the COVID-19 lockdown. However, However, the ECPs reported lack of confidence on the non-inclusion of eye care workers as essential in the government guidelines for the control of this pandemic, which places them at increased risk. To therefore ensure that ECPs continue to provide the needed services during the pandemic or similar events, there is need for training on the proper use of PPE and recognition as essential worker, this will in turn boost their confidence when attending to patients even after the lockdown. The Nigerian government need to strengthen health systems by improving and extending training on standard infection prevention and control measures for effective control of the pandemic.
Reviewer 2 Report
Review Comments
Manuscript ID: ijerph-862474
Type of manuscript: Article
Title: Assessment of Knowledge, Practice and Guidance towards the Novel
COVID-19 among Eye Care Practitioners in Nigeria – A survey based study
Submitted to section: Environmental Health
This research is valuable to provide immediate information to the global large-scale epidemic! Important reference information for less-developed and developing countries to develop strategies for immediate response.
Major comments
This is a very valuable article, which can quickly and instantly seize the study chance in the period of the COVID-19 epidemic and provide immediate and precious medical and public health information. Online survey was distributed to explore the knowledge, practice of risk and guidelines of the COVID-19 infection among the eye care practitioners and the potential associated factor in Nigeria, between 2nd - 8th May 2020 corresponding to the week of the lockdown in to eye care practitioners. 823 data were analyzed with univariate and multivariate logistic regression to examine the associated factors. While the medical practitioners having more knowledge from better trainings, for the good sake of majority the governments need to strengthen health systems by improving and extending training on standard infection prevention and control measures for effective control of the pandemic.
The paper is accepted. No suggestions for reversion, here.
Author Response
Reviewer 2
This research is valuable to provide immediate information to the global large-scale epidemic! Important reference information for less-developed and developing countries to develop strategies for immediate response.
Major comments
This is a very valuable article, which can quickly and instantly seize the study chance in the period of the COVID-19 epidemic and provide immediate and precious medical and public health information. Online survey was distributed to explore the knowledge, practice of risk and guidelines of the COVID-19 infection among the eye care practitioners and the potential associated factor in Nigeria, between 2nd - 8th May 2020 corresponding to the week of the lockdown in to eye care practitioners. 823 data were analysed with univariate and multivariate logistic regression to examine the associated factors. While the medical practitioners having more knowledge from better trainings, for the good sake of majority the governments need to strengthen health systems by improving and extending training on standard infection prevention and control measures for effective control of the pandemic.
Response:
Thanks for the useful comments. We have also included your suggestion in the conclusion.
“The Nigerian government need to strengthen health systems by improving and extending training on standard infection prevention and control measures for effective control of the pandemic.”
Reviewer 3 Report
A very interesting article addressing current issues of 2020, evaluating the knowledge, attitudes and guidance of eye care 255 practitioners (ECPs) in Nigeria regarding the Public Health initiatives for the novel coronavirus. It is well-written and well-composed. All tables and figures are presented. Besides the fact that they significantly simplify the text, they are well designed and easy to percept. Introduction is comprehensive and provides a sufficient background to interest the reader and make him keep reading. The authors critically take their results. Sample size is sufficient and correctly calculated.
The only grammatical mistake presented:
page 7 line 227-228 the way to deliver
Author Response
Reviewer 3
A very interesting article addressing current issues of 2020, evaluating the knowledge, attitudes and guidance of eye care 255 practitioners (ECPs) in Nigeria regarding the Public Health initiatives for the novel coronavirus. It is well written and well composed. All tables and figures are presented. Besides the fact that they significantly simplify the text, they are well designed and easy to percept. Introduction is comprehensive and provides a sufficient background to interest the reader and make him keep reading. The authors critically take their results. Sample size is sufficient and correctly calculated.
The only grammatical mistake presented:
page 7 line 227-228 the way to deliver
Response:
Thanks for the useful comments. We could not find any section on ‘the way to deliver’
Round 2
Reviewer 1 Report
The article has been significantly improved and could be published.